# Generative Modeling of Individual Behavior at Scale

## Abstract

Recent years have seen a growing interest in using AI to model human behavior, particularly in domains where humans learn from or collaborate with this technology. While most existing work attempts to model human behavior at an aggregate level, our goal is to model behavior at the individual level. Recent work in the domain of chess has shown that *behavioral stylometry*, or the task of identifying a person from their actions alone, can be achieved with high accuracy among a pool of a few thousand players. We provide a new perspective on behavioral stylometry by connecting it to the vast literature of transfer learning in NLP. Specifically, by casting the stylometry problem as a multi-task learning problem—where each task represents a distinct *person*—we show that parameter-efficient fine-tuning (PEFT) methods can be adapted to perform stylometry at an unprecedented scale (47,864 players), while enabling few-shot learning for unseen players. Our approach leverages recent modular PEFT methods to learn a shared set of skill parameters that can be combined in different ways via *style vectors*. Style vectors enable two important capabilities. First, they are generative: we can generate actions in the style of a player simply by conditioning on the player's style vector. Second, they induce a latent style space that we can interpret and manipulate algorithmically. This allows us to compare different player styles, as well as synthesize new (human-like) styles, e.g. by interpolating between the style vectors of two players.

## 1 Introduction

With the rapidly advancing capabilities of machine learning in recent years, it has become increasingly important to find constructive ways for humans to interact with this technology. Even in domains where AI has achieved proficiency or superhuman ability, there is still considerable value in using machine learning to understand how humans approach these tasks. Such an understanding can help identify areas for improvement in humans, find compatible human partners, develop effective AI collaborators or teachers, and create more human-like experiences, for example.

A common method for capturing human behavior is behavioral cloning (BC), a form of imitation learning (Schaal, 1996) which applies supervised learning on fixed demonstrations collected for a given task. While traditionally used in domains such as robotics (Florence et al., 2022) and self-driving vehicles (Pomerleau, 1988), BC has seen increasing use in gaming, where the above use cases can be explored in a safe, virtual environment. Indeed, recent work in games like Counter-Strike (Pearce & Zhu, 2022; Pearce et al., 2023) and Overcooked (Carroll et al., 2019; Strouse et al., 2021; Yang et al., 2022) have developed human-like models that match human demonstrations closely; in the case of Overcooked, these models were used to train AI partners that collaborate effectively with human players. While the majority of this work has focused on modeling human behavior in aggregate, we believe the most value can be derived from modeling human behavior at the individual level. To that end, recent results in the game of chess have shown the most promise. McIlroy-Young et al. (2020) used behavior cloning to create a set of models called Maia that match human play at 9 aggregate skill levels. By fine-tuning these models on the data of 400 individual players, they created 400 personalized models that achieve a 4-5% higher move-matching accuracy on average compared to the best performing aggregate Maia model (McIlroy-Young et al., 2022). The authors show that this is sufficient to perform *behavioral stylometry* with high accuracy, where the goal is to identify which person played a given query set of games. In this case, the authors simply apply the 400 personalized models to the query set and output the model with the highest

move-matching accuracy. As a more scalable approach, the authors designed a Transformer-based embedding for games and players (a player's embedding is the average of their game embeddings), and showed that this can be used to perform stylometry among 2,844 players with high accuracy, by embedding the query set of games and matching it to the closest player in the embedding (McIlroy-Young et al., 2021). This method supported few-shot learning of unseen players given a reference set of 100 games. Unlike the personalized models, however, the embedding was not generative: it could be used to identify players, but could not generate moves in their playing style.

This paper advances the above work in three important ways, focusing on the game of chess:

1. We perform behavioral stylometry at an unprecedented scale, across 47,864 players. Using query sets of 100 games, we achieve an accuracy of 94.4%.
2. Our model is generative: it learns an explicit *style vector* for each player that can be used to generate actions in their playing style. Our per-player models have move-matching accuracy in the range 45-69%, with a mean of 59%, even for players with very few (e.g., 50) games.
3. Our style vectors induce a latent space that can be interpreted and manipulated algorithmically. This allows us to analyze a player's style using human-understandable concepts, and synthesize new, human-like styles by interpolating between two existing player styles.

Our key insight for achieving these results is that behavioral stylometry can be viewed as a multi-task learning problem, where each *task* represents an individual *person*. The goal here is to generalize across an initial set of players (tasks) while supporting few-shot learning of new players (tasks). To do this efficiently, we leverage recent advances in parameter-efficient fine-tuning (PEFT) (Ponti et al., 2023; Caccia et al., 2022). Specifically, we augment an existing model (such as Maia) with a set of Low Rank Adapters (LoRAs) and a routing matrix that specifies a distribution over these adapters for each player. Unlike approaches that train a separate LoRA for each task, this modular design allows players to softly share parameters in a fine-grained manner. We apply this adapter framework to a modified version of the Maia model. By training Maia to convergence across all player data *before* fine-tuning the adapters and routing matrix on per-player data, our methodology encourages the adapters to learn different facets of knowledge, or *latent skills*, that explain the variance between players. Since each row of the routing matrix implies a distribution over these latent skills, we can interpret each row as a *style vector* for the corresponding player. Our experiments show that this vector is a good characterization of the player's style, enabling accurate stylometry at scale, and facilitating interpretation and synthesis of styles.

## 2 BACKGROUND

In multitask learning (Caruana, 1997; Ruder et al., 2019), we are given a collection of tasks $\mathcal{T} = (\mathcal{T}_1, \ldots, \mathcal{T}_{|\mathcal{T}|})$, each task $\mathcal{T}_i$ associated with a dataset $\mathcal{D}_i = \{(x_1, y_1), ..., (x_{n_i}, y_{n_i})\}$. Multitask learning exploits the similarities among related training tasks by transferring knowledge among them, which in turn increases sample efficiency. Ideally, it builds representations easily adaptable to new tasks from potentially few target examples. The premise of this paper is that modeling individual human behavior from a pool of players can be interpreted as a multitask learning problem. In other words, each task $\mathcal{T}_i$ consists of modeling the behavior of a specific player; and the dataset $\mathcal{D}_i$ corresponds to the set of game moves made by player $i$. Specifically, an $(x, y)$ tuple denotes a chess board position at a specific point in time during a game, along with player $i$'s move in this game state. For the rest of the paper, we will thus use the notion of *tasks* and *players* interchangeably.

### 2.1 PARAMETER-EFFICIENT FINE-TUNING

Popularized in NLP, parameter-efficient fine-tuning (PEFT) (Houlsby et al., 2019; Hu et al., 2022; Liu et al., 2022) approaches have emerged as a scalable solution for adapting Large Language Models to several downstream tasks. Indeed, standard finetuning of pretrained LLMs requires updating possibly billions of parameters for each task at hand, which makes storing and serving each task-specific model expensive. PEFT methods instead freeze the pretrained model and inject a small set of trainable task-specific weights, or "adapters".

One such approach is the use of Low Rank Adapters (LoRA) (Hu et al., 2022), which modify linear transformations in the network by adding a learnable low rank shift

$$h = \left(\boldsymbol{W}_0 + \Delta\boldsymbol{W}\right) x = \left(\boldsymbol{W}_0 + \boldsymbol{A}\boldsymbol{B}^T\right) x. \tag{1}$$

Here, $\boldsymbol{W}_0 \in \mathbb{R}^{d \times d}$ are the (frozen) weights of the pre-trained model, and $\boldsymbol{A}, \boldsymbol{B} \in \mathbb{R}^{d \times r}$ the learnable low-rank parameters of rank $r \ll d$. With this approach, practitioners can trade off parameter efficiency with expressivity by increasing the rank $r$ of the transformation.

## 2.2 POLYTROPON AND MULTI-HEAD ADAPTER ROUTING

Standard PEFT methods such as LoRA can adapt a pretrained model for a given task. In multitask settings, a common approach is to train individual sets of adapters for each task. This approach however is suboptimal, as it does not enable any sharing of information, or *transfer*, across similar tasks. On the other hand, if one uses the same set of adapters across all tasks, they risk *negative interference* (Wang et al., 2021) across dissimilar tasks, which may harm optimization and performance. To this end, Polytropon (Ponti et al., 2019) (`Poly`) addresses this transfer / interference tradeoff by softly sharing parameters across tasks.

That is, each `Poly` layer contains 1) an inventory of LoRA adapters

$$\mathcal{M} = \{\boldsymbol{A}^{(1)}\boldsymbol{B}^{(1)}, \dots, \boldsymbol{A}^{(m)}\boldsymbol{B}^{(m)}\},$$

with $m \ll |\mathcal{T}|$ and 2) a task-routing matrix $\boldsymbol{Z} \in \mathbb{R}^{|\mathcal{T}| \times m}$, where $\boldsymbol{Z}_\tau \in \mathbb{R}^m$ specifies the task $\tau$'s distribution over the shared modules. This formulation allows similar tasks to share the same adapters and yield transfer, while still being flexible so that dissimilar tasks may have a non-overlapping set of parameters. The collection of adapters $\mathcal{M}$ can be interepreted as capturing different facets of knowledge, or *latent skills*, of the full multitask distribution.

At each forward pass, `Poly` LoRA adapters for task $\tau$ are constructed as follows:

$$\boldsymbol{A}^\tau = \sum_i \alpha_i \boldsymbol{A}^{(i)}; \; \boldsymbol{B}^\tau = \sum_i \alpha_i \boldsymbol{B}^{(i)} \tag{Poly}$$

where $\alpha_i = \texttt{softmax}(\boldsymbol{Z}[\tau])_i$ denotes the mixing weight of the $i$-th adapter in the inventory, and $\boldsymbol{A}^{(i)}, \boldsymbol{B}^{(i)}, \boldsymbol{A}^\tau, \boldsymbol{B}^\tau \in \mathbb{R}^{d \times r}$. Here, the $\tau$-th row of the routing matrix $\boldsymbol{Z}$ is effectively selecting which adapter modules to include in the linear combination of the adapter for task $\tau$. In the stylometry setting, where each task consists of modeling an individual, $\boldsymbol{Z}[\tau]$ specifies which latent skills are activated for user $\tau$. As per Eqn 1, the final output of the linear mapping modified with a `Poly` LoRA adapter becomes $h = \left(\boldsymbol{W}_0 + \boldsymbol{A}^\tau(\boldsymbol{B}^\tau)^T\right) x$.

In `Poly`, the module combination step remains *coarse*, as only linear combinations of the existing modules can be generated. Caccia et al. (2022) propose a more fine-grained module combination approach, referred to as Multi-Head Routing (`MHR`). Similar to Multi-Head Attention (Vaswani et al., 2017), the input dimension of $\boldsymbol{A}$ (and output dimensions of $\boldsymbol{B}$) are partitioned into $h$ heads, where a `Poly`-style procedure occurs for each head. The resulting parameters from each head are then concatenated, recovering the full input (and output) dimensions. See Appendix A.1 for a detailed description of `MHR`.

**Routing-only fine-tuning** While LoRA adapters can reduce the parameter cost from billions to millions (Liu et al., 2022), deploying a set of adapters for each new task can still be prohibitive when dealing with thousands of tasks. To this end, Caccia et al. (2022) proposed routing-only finetuning (`MHR`-$z$), where after an initial phase of pretraining, the adapter modules are fixed, and only the routing parameters $\boldsymbol{Z}$ are learned for a new task. This reduces the parameter cost for each additional task by several orders of magnitude, while performing similarly as standard adapter training.

## 3 METHODOLOGY

In this section, we detail our methodology for creating a generative model of individual player behavior that supports stylometry, style analysis, and style synthesis. We start by applying the `MHR` adapter framework to a modified version of Maia's architecture, and then discuss our model training and evaluation methodology, as well as our approach for analyzing and synthesizing style.

### 3.1 MODEL ARCHITECTURE

The base model used for our experiments is the Squeeze-and-Excitation Residual Network proposed in Hu et al. (2018). This approach explicitly models the interdependencies across channels. At every residual block, channel information is aggregated across spatial dimensions via a global pooling

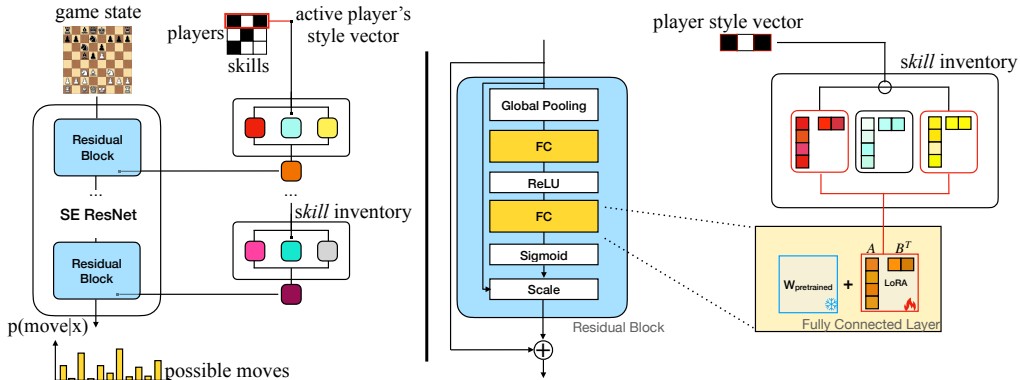

Figure 1: (left) Overall MHR-Maia architecture. We modify the base Maia model to predict a chess move in the style of an individual player by inserting adapters generated from the player's style vector. (right) Inside each residual block, we inject a low-rank adapter at every fully-connected layer. This adapter is generated from a skill inventory that is shared across players. The player's style vector specifies which skills are active to generate the player's adapter. In this example, following the active player's style vector, the first and third skill in the inventory are selected and averaged, and the resulting low rank weight shift is applied on the (frozen) fully connected layer.

operation. The resulting vector is then processed by a two layer MLP, with a bottleneck representation compressing the number of channels by $r$. The output of this MLP is a one-dimensional vector used to scale the output of the residual block along the channel dimension. The workflow of an SE ResNet layer is shown in Fig. 1 (left). In practice, we use 12 residual blocks containing 256 filters, and a bottleneck compression factor of $r = 8$. We note that this differs from the base Maia model in McIlroy-Young et al. (2022), which uses 64 filters and 6 residual blocks.

To enable user-based adaptation, we incorporate the MHR adapters described in Sec. 2.2 into our base model. For every linear transformation in the two-layer MLP used for the channelwise rescaling operation described above, we add an MHR layer built of LoRA adapters. In our experiments, the size of our inventory adapter is 32, and we use a multi-head routing strategy with 8 heads. Therefore, for each user we must learn $32{\times}8{=}256$ routing parameters, and these parameters make up our player style vector. To facilitate interpretability and style analysis, we use the same routing (or style vector) across all MHR layers. A depiction of this framework is shown in Fig. 1(right). We used a rank of 16 for each LoRA, across all $12{\times}2{=}24$ MHR layers, totaling 5M additional parameters.

## 3.2 DATA COLLECTION

We use data from the largest open-source online chess platform, Lichess.org (Duplessis, 2021), which boasts over 100 million games played per month, a globally diverse player community, and a database of over 4.8 billion games. We collected Blitz games played between 2013 and 2020 inclusive—these are games with 3 or 5 minutes per side, optionally with a few seconds of time increment per move—and applied the same player filtering criteria as McIlroy-Young et al. (2022). The resulting dataset comprises 47,864 unique players and over 244 million games. We enforce a strict lower bound of 10 games per player to preserve validation and testing sets, though most players tend to have many more games (e.g. the median player has 3,479 games). See Appendix A.2 for a discussion on data imbalance. We divide this set of players into a few subsets to support our training methodology below. The *base player* set is used to train the base Maia model. The *fine-tuning player* set is used to fine-tune the MHR-Maia architecture shown in Fig. 1. The *few-shot player* set is used for few-shot learning on the MHR-Maia. For players used during base model training or MHR fine-tuning, we split their games into a training set (80%), validation set (10%), and testing set (10%). For players used during few-shot learning, we use a *reference set* of 100 games to learn their style vectors, and a *query set* of 100 games to evaluate them for stylometry. If a few-shot player was previously trained on during fine-tuning, we refer to them as *seen*; otherwise, they are *unseen*. Finally, to enable a direct comparison with the personalized Maia model-based (McIlroy-Young et al., 2022) and embedding-based (McIlroy-Young et al., 2021) stylometry approaches, we create a fine-tuning player set consisting of the same 400 players used in those studies (with the same games used in the training/testing sets).

Currently, we treat each player's data holistically, but in principle one could partition a player's data in different ways to perform a finer analysis of their playing style. We explore this in Appendix A.4.

### 3.3 MODEL TRAINING AND EVALUATION

**Base model.** We train our (modified) base Maia model using data from a base player set of all 47,864 players, treating this as a classification task of predicting human move $y$ made in chess position $x$, given a datapoint $(x, y)$. We use the Adam optimizer (Kingma & Ba, 2015) with a learning rate of 0.0003 (and no learning rate schedule). We use the same loss functions and evaluation criteria as the original Maia work: Maia's policy head uses a cross entropy loss while the value head uses MSE; the output of the policy head is used to evaluate the model's move-matching accuracy. Specifically, given data from the testing sets of all players, we compare the move with highest probability from the policy head with the ground truth label (the human move).

**MHR fine-tuning.** To fine-tune the MHR-Maia architecture shown in Fig. 1, we adopt the methodology used in Caccia et al. (2022), namely, we *freeze the Maia model* and fine-tune the MHR layers and routing matrix using data from a fine-tuning player set. Recall that the routing matrix $Z$ has a row (style vector) for each player in the fine-tuning set. When fine-tuning on data from player $i$, Maia's output is influenced during the forward pass through the adapter modules activated by player $i$'s style vector. During the backward pass, the gradient updates flow through the shared adapters but only update player $i$'s style vector. Our fine-tuning process uses the Adam optimizer with a higher learning rate of 0.005 for the LoRAs, and an even higher learning rate of 0.02 for the style vectors.

We use two fine-tuning player sets in our experiments, creating two separate MHR-Maia models. The first set comprises all 47,864 players and is used to evaluate behavioral cloning and stylometry at very large scale. The second set is comprised of the same 400 players used by McIlroy-Young et al. (2022); we use this to evaluate few-shot learning and stylometry on unseen players, and compare these results to the original Maia work.

We evaluate a fine-tuned MHR-Maia model in two ways. First, we measure its move-matching accuracy, similar to how we evaluate the base Maia model. However, since MHR-Maia provides a generative model for each player (which can be invoked by indexing into the player's style vector), we can separately evaluate each player's move-matching accuracy by invoking their generative model and evaluating it on the player's testing set. The overall move-matching accuracy for the model is simply the average of these per-player accuracies.

Our second evaluation method uses the model to perform behavioral stylometry among all players in the fine-tuning set. In theory, we could adopt the methodology of McIlroy-Young et al. (2022) and compute the move-matching accuracy of every player applied to every other player's query set, but such a quadratic computation is infeasible beyond a few thousand players. A much more efficient approach is to simply compare the style vectors. That is, given a query set of games, we learn a new style vector in $Z$ for those games via few-shot learning (discussed below), and compare this vector to every other vectors in $Z$. Using cosine similarity as our distance measure, we then output the the player with the highest cosine similarity to the query set vector.

**Few-shot learning.** To perform few-shot learning on MHR-Maia, we perform "routing-only fine-tuning" as described in section 2.2 and *additionally freeze all MHR adapters*, allowing only the entries of $Z$ to change. Specifically, given a reference set of games for a few-shot player, we add a (randomly-initialized) new row to $Z$ and fine-tune it on data from the reference set, eventually learning a style vector for the player. Using this style vector, we can invoke a generative model of the player and use it to evaluate move-matching accuracy, as described above. To perform stylometry, if the player is a seen player (i.e., part of the fine-tuning set), then a matching style vector already exists in $Z$ and we can find it using cosine similarity, as described above. Otherwise, if the player is unseen, then we simply repeat the few-shot learning process on a query set of games (from the same player), and compare this new style vector to the entries in $Z$. All of our few-shot experiments use the MHR-Maia model fine-tuned on the 400-player set from McIlroy-Young et al. (2022).

### 3.4 STYLE ANALYSIS AND SYNTHESIS

The style vectors in $Z$ represent distinct distributions over latent skills that give us a starting point for comparing player styles. For example, we can use the cosine similarity of these vectors to determine how similar or different players are, as employed by our stylometry methodology above.

In order to make such comparisons more human-understandable, we exploit the generative nature of MHR-Maia to connect style vectors to human concepts in chess. Specifically, inspired by the concept probing techniques used to analyze AlphaZero (a deep RL chess engine) (McGrath et al., 2022), we use a set of human-coded heuristic functions found in Stockfish (a traditional chess engine) that are used to evaluate a given chess position. These functions capture concepts such as: king safety, material imbalance, piece mobility, and so on. By invoking each player's generative model on a fixed set of chess positions and seeing which move they select, we can measure the deltas in the heuristic functions before and after the move, and use this to summarize how much emphasis each player places on the corresponding concepts.

In addition to interpreting existing style vectors, we can also synthesize new ones. We do this by performing simple vector arithmetic, for example by interpolating between two players using a convex combination of their style vectors. This allows us to create new player styles that are human-like and yet previously unseen in the world. To determine the playing strength of these new players, we run a simulated tournament between them and the players they are derived from. The results of this tournament can be used to calculate a win probability (based on a standard scoring system of win=1 point, loss=0 points, draw=0.5), which can then be converted to an estimated Elo rating.

## 4  EXPERIMENTS

In this section, we demonstrate two main findings. First, MHR-Maia performs competitively with previous methods in their respective tasks at unprecedented scale. We are able to closely match the accuracy of move prediction with individual models (McIlroy-Young et al., 2022), while also retaining the computational efficiencies of stylometry with a Transformer-based method (McIlroy-Young et al., 2021). Second, we show that explicitly capturing style vectors allows us to reason about and perform arithmetic operations on generated behaviors. Specifically, we generate new players by combining learned style vectors in a way similar to how early vector-based image generation showed the ability to generate image analogies (Radford et al., 2016).

### 4.1  MOVE GENERATION

Here we compare the efficacy of our method to using individually fine-tuned models for each player. Fine-tuning individual models generally results in superior results compared to PEFT methods, as the increased parameter count produces more expressive models. However, they are also more computationally intensive to train and store. That said, in the domain of modeling individual behavior in chess, MHR-Maia is able to perform comparatively well despite using a much smaller parameter budget. Figure 2 shows that MHR-Maia matches individual model fine-tuning over a wide range of game counts. For 25, 50, and 100 games, the LoRAs are frozen to combat overfitting (Table 1), and only player style vectors are fit (256 parameters). The base model is frozen for all game counts in MHR-Maia. Table 1 details move-matching performance with various parameters. The model has already learned the set of skills required to differentiate the players, all that is needed with very few-shot learning is to find a proper recombination of the learned skills within the new style vectors. The Transformer-based method is omitted, as it is incapable of generating moves.

### 4.2  BEHAVIORAL STYLOMETRY

In this section, we show that our models perform competitively with previous behavioral stylometry methods for both seen and unseen players. Here, the goal is to predict the player who produced a given set of games. We compare to individual model fine-tuning (McIlroy-Young et al., 2022), fitting a pre-trained Maia to the data from a single player, and to a Transformer-based method (McIlroy-Young et al., 2021), which embeds players in a 512-dimensional style space based on their gameplay. All reported accuracies are top-1 unless stated otherwise.

To perform stylometry on a query set of games, McIlroy-Young et al. (2022) suggest measuring the move-matching accuracy of each available fine-tuned model and selecting the one that seems most performant. As seen in Table 2, this procedure works well, but is tremendously expensive—requiring computationally intensive inference calls on the entire query set for every candidate player.

In contrast, both the Transformer-based method and MHR-Maia scale much better to large numbers of players. The Transformer-based method needs only to condition on these games to produce a vector, while MHR-Maia needs only to fit a new vector. In either case, the produced vectors need

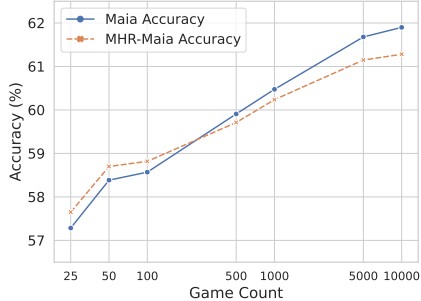

| Players | Games | Acc. (%) | Few-Shot |
|---------|-------|----------|----------|
| 50 | 50 | 58.1 | ✗ |
| 50 | 1000 | **60.4** | ✗ |
| 50 | 50 | **58.7** | ✓ |
| 50 | 1000 | 59.2 | ✓ |
| 10000 | 100 | 58.1 | ✓ |
| 400 | – | 61.5 | ✗ |
| 47864 | – | 59.0 | ✗ |
| Base Model | – | 54.4 | – |

Figure 2: Accuracy at various game counts of the individual models (Maia) and our method (`MHR-Maia`). For 100 games and fewer, the LoRA modules in `MHR-Maia` are frozen.

Table 1: Move prediction accuracy with various game counts. Few-Shot indicates whether LoRA modules are frozen. Unreported game counts vary per player.

| Method | $|Query|$ | $|Universe|$ | $|Query\ Games|$ | Random (%) | Acc. (%) |
|--------|-----------|--------------|------------------|------------|----------|
| McIlroy-Young et al. (2022) | 400 | 400 | 100 | 0.25 | 98.0 |
| McIlroy-Young et al. (2021) | 400 | 400 | 100 | 0.25 | 99.5 |
| `MHR`-Maia | 400 | 400 | 100 | 0.25 | **99.8** |
| `MHR`-Maia | 10000 | 47864 | 100 | 0.002 | **94.4** |
| McIlroy-Young et al. (2022) | 400 | 400 | 30 | 0.25 | 94.0 |
| `MHR`-Maia | 400 | 400 | 30 | 0.25 | **98.8** |
| `MHR`-Maia (100 games) | 10000 | 10000 | 100 | 0.01 | 87.6 |
| McIlroy-Young et al. (2021) | 578 | 2844 | 100 | 0.035 | 79.1 |

Table 2: (Top) Stylometry accuracy on seen users. (Bottom) Stylometry accuracy on unseen users, after initial training on the 400-player dataset (McIlroy-Young et al., 2022). Numbers for McIlroy-Young et al. (2022) and McIlroy-Young et al. (2021) are borrowed from their respective papers.

only be matched to those in the player set, e.g., using cosine similarity. Table 2 compares both approaches, showing that `MHR`-Maia performs competitively or better, on a much larger universe. To do this, we use few-shot learning to compute style vectors for 10,000 players based on their 100 game reference sets, then fit new style vectors for each player based on their respective query sets. Note that the individual model fine-tuning method is omitted from the larger few-shot study due to scalability reasons. The Transformer-based method can scale, but it is not a generative model.

### 4.3 ANALYSIS AND SYNTHESIS OF STYLE VECTORS

This section provides evidence for the consistency of style vectors in `MHR`-Maia and probe the information learned within them. We additionally create novel intermediary styles of players. Finally, we visualize the individual style of our vectors using traditional chess evaluation methods.

**Style vector consistency.** The result shown in Figure 3 suggests that `MHR`-Maia, used to fit player vectors, will find similar vectors for two non-overlapping datasets if they are generated by the same player. Here we examine that phenomenon in more detail. Figure 3 is created by splitting 50 players data into an average of 50 separate game sets each. We then train each game set as a separate player, and measure to cosine similarity between vectors that are learned from the same player to those learned by different players. We find that the cosine similarity between same-player vectors is far larger than those measured between the 47,864 different players, demonstrating a self-consistency in what is captured by these vectors. To further probe the consistency of this style space, we create intermediary players by merging two players' game sets. Interestingly, we find that the vectors fitted to these joint datasets are quite similar to vectors obtained by simply averaging the vectors of each of the distinct player datasets. This Figure 4 shows how these cosine similarities are distributed.

**Relative player quality.** Learned style vectors appear to capture information related to the quality of the corresponding gameplay. For example, these vectors contain information related to Elo score, the standard system for ranking the relative quality of players. A linear regressor trained to predict

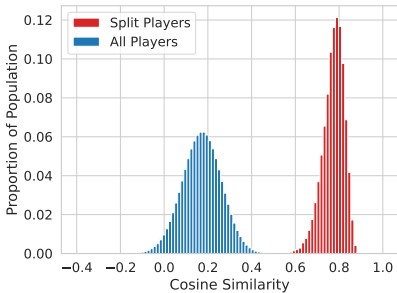 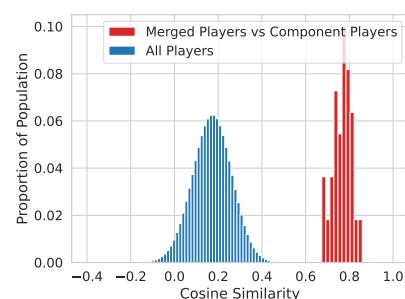

Figure 3: Cosine similarity between the cosine similarity of vectors learned with split game sets compared to the population of 48k players.

Figure 4: Cosine similarity between merged vectors and their component vectors compared to the population of 48k players.

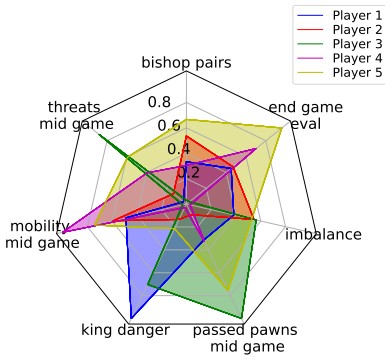 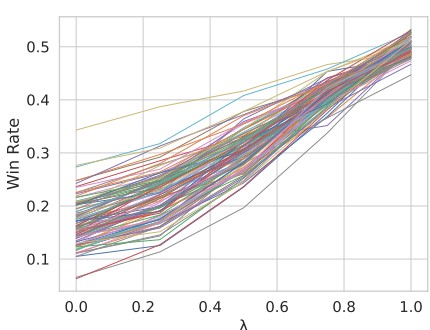

Figure 5: Comparing player styles using human-interpretable evaluation metrics.

Figure 6: Winrate as a weaker player is interpolated with a stronger player as a function of $\lambda$.

player Elo from their style vector is able to do so with a mean absolute error (MAE) of 54—a naive approach that simply picks the mean Elo obtains an MAE of roughly 204. This experiment was carried out over a test set of 7,864 randomly held-out players with an average Elo rating of 1600.

**Differences in player strategy.** To confirm the distinctness of playstyle across style vectors, we sampled five learned player vectors uniformly at random from a `MHR`-Maia model trained on the 400 players from McIlroy-Young et al. (2022). For each, we predicted their preferred move across $2^{17}$ roughly neutral board positions (within 1 pawn of balanced). For each move, we measured a series of Stockfish evaluation metrics, which are human-interpretable scores for different qualities of chess gameplay, including piece mobility, king safety, etc. Figure 5 shows the distribution of these metrics for each of the five players, demonstrating that these vectors store a wide diversity of styles.

**Convex combinations.** Finally, we show that interpolating between skill vectors results in a player whose level is a weighted average of the interpolated players. Here, we take 100 pairs of learned player vectors, such that one item in the pair corresponds to a strong player and the other to a weaker player. We then gradually interpolate between the weak and strong player as $(1 - \lambda)u_w + \lambda u_s$, $0 \leq \lambda \leq 1$, where $u_w$ and $u_s$ are respectively vectors representing the weak and strong player. For each value of $\lambda$ we simulate 1,000 games between the interpolated vector and $u_s$, the stronger player. Figure 6 plots the win rate of the interpolated player as a function of $\lambda$ for each player pair we considered. This plot demonstrates that win rate progresses in a roughly linear fashion, starting off winning infrequently against the stronger player and eventually winning roughly half the time as the interpolated player converges to the stronger player. In Appendix 6, we further investigate the effect of interpolation on style using the the human-interpretable Stockfish evaluation metrics. These results suggest that simple arithmetic operations in this vector space seem to respect human-identifiable notions of style.

## 5 RELATED WORK

We discuss related work on stylometry, player style modeling (outside of chess), as well as other PEFT methods for transfer and multi-task learning.

### 5.1 STYLOMETRY AND PLAYER STYLE MODELING

Originally referring to performing author attribution via statistical analysis of text (Tweedie et al., 1996; Neal et al., 2017), stylometry has since come to refer to the general task of identifying individuals given a set of samples or actions and has found broad application for tasks such as handwriting recognition (Bromley et al., 1993), speaker verification (Wan et al., 2018), identifying programmers from code (Caliskan-Islam et al., 2015), determining user age and gender from blog posts (Goswami et al., 2009), and identifying characteristics of authors of scientific articles (Bergsma et al., 2012). Along these lines, McIlroy-Young et al. (2021) introduced behavioral stylometry, the task of identifying individuals based on their decisions or actions in a given situation. In the context of games (such as chess), this takes the form of identifying players based on their in-game actions or moves. This characterization can be viewed as a form of playstyle modeling, where an individual's playing style is defined by the actions they choose to perform in a certain game state (Holmgård et al., 2013). This has been used to determine a player's style by building agents representative of different playstyles and then assigning a player the style of the agent whose behavior most closely matches the player (Holmgård et al., 2014). An alternate approach for identifying playstyles involves gathering gameplay data (e.g., event logs, player trajectories) and applying methods such as clustering (Ingram et al., 2022), LDA (Gow et al., 2012), bayesian approaches (Normoyle & Jensen, 2015) and sequential models (Valls-Vargas et al., 2015) to identify groups of players with similar styles. Note that, unlike our work, these approaches do not learn a behavior model conditioned on the learned styles to generate actions suited to a specific style. Also worth noting is that both prior sets of approaches focus on identifying aggregate playstyles. In a similar vein, behavior cloning methods for games also typically learn models of behavior aggregated across a number of individuals as seen in such approaches for games like Counter-Strike (Pearce & Zhu, 2022) and Overcooked (Carroll et al., 2019). Our work differs in that we model behavior separately for each individual player.

### 5.2 PARAMETER-EFFICIENT ADAPTATION

Approaches for efficient adaption of a pretrained model can be broadly grouped in two categories. Adapter based methods inject new parameters within a pretrained model, and only updates the newly inserted parameters while keeping the backbone fixed. Houlsby et al. (2019) defines an adapter as a two-layer feed-forward neural network with a bottleneck representation, and are inserted before the multi-head attention layer in Transformers. Similar approaches have been used for cross-lingual transfer (Pfeiffer et al., 2020). Adapters have also been used in vision based multitask settings (Rebuffi et al., 2017). More recently, Ansell et al. (2022) propose to learn sparse masks, and show that these marks are composable, enabling zero-shot transfer. Lastly, Hu et al. (2022) learn low-rank shifts on the original weights, and (Liu et al., 2022) learns an elementwise multiplier of the pretrained model's activations. Adapters have also been used in multitask settings. Chronopoulou et al. (2023) independently trains adapters for each task. In order to transfer to new tasks, the authors merge the parameters of the adapters of relevant training tasks.

Another popular approach is the use of soft prompts (Lester et al., 2021). In the context of natural language, where the input is a sequence of work tokens, soft prompts append new, learnable tokens to the sequence. Closer to the multitask setting presented in this work, Vu et al. (2021) learns a collection of soft-prompts amassed from a multitask training set, and given a novel task, retrieves relevant prompts for efficient transfer.

## 6 CONCLUSION

We show that behavioral cloning of individuals is possible at very large scale in the game of chess. We cast this problem in the framework of multi-task learning and employ modular parameter-efficient fine-tuning methods to learn a shared set of skills across players, modulated by a distinct style vector for each player. We use these style vectors to perform stylometry, predict Elo ratings, probe a player's playing style, and create novel playing styles.

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

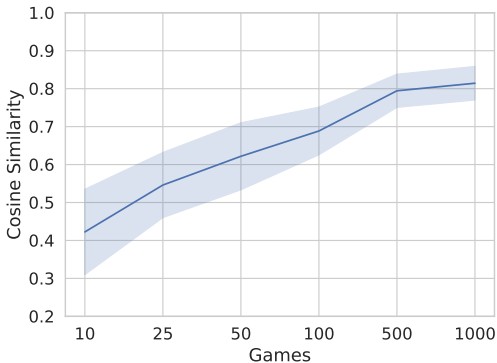

Figure 7: Cosine Similarity of vectors of 50 players trained with varying game sizes to a vector trained with 10,000 games.

## A APPENDIX

### A.1 MULTI-HEAD ADAPTER ROUTING

In `Poly`, the module combination step remains *coarse*, as only linear combinations of the existing modules can be generated. Caccia et al. (2022) propose a more fine-grained module combination approach, referred to as Multi-Head Routing (`MHR`). Similar to Multi-Head Attention (Vaswani et al., 2017), the input dimension of $\boldsymbol{A}$ (and output dimensions of $\boldsymbol{B}$) are partitioned into $h$ heads, where a `Poly`-style procedure occurs for each head. The resulting parameters from each head are then concatenated, recovering the full input (and output) dimensions. This makes the module combination step *piecewise linear*, with a separate task-routing matrix $\boldsymbol{Z}$ learned for each head.

Formally, a `MHR` layer learns a 3-dimensional task-routing tensor $\mathsf{Z} \in \mathbb{R}^{|\mathcal{T}| \times |\mathcal{M}| \times h}$. The 2D slice $\mathsf{Z}_{:,:,k} \in \mathbb{R}^{|\mathcal{T}| \times |\mathcal{M}|}$ of the tensor $\mathsf{Z}$ denotes the distribution over modules for the $k$-th head, and $\boldsymbol{W}[k] \in \mathbb{R}^{\frac{d}{h} \times r}$ the $k$-th partition along the rows of the matrix $\boldsymbol{W} \in \mathbb{R}^{d \times r}$. The adapter parameters $\boldsymbol{A}^{\tau} \in \mathbb{R}^{d \times r}$ for task $\tau$, and for each adapter layer, are computed as (similarly for $\boldsymbol{B}^{\tau}$):

$$\boldsymbol{A}_k^{\tau} = \sum_j \alpha_{i,k} \cdot \boldsymbol{A}_j[k] \ \text{ with } \ \boldsymbol{A}_k^{\tau} \in \mathbb{R}^{\frac{d}{h} \times r}, \quad\quad\quad \text{(MHR)}$$
$$\boldsymbol{A}^{\tau} = \texttt{concat}(\boldsymbol{A}_1^{\tau}, \ldots, \boldsymbol{A}_h^{\tau}),$$

where $\alpha_{i,k} = \texttt{softmax}(\boldsymbol{Z}[\tau,:,\texttt{k}])_i$. Importantly, the number of LoRA adapter parameters does not increase with the number of heads. Only the task-routing parameters linearly increase with $h$ for `MHR` vs. `Poly`. However, this cost is negligible as the parameter count of the routing matrices is much smaller than for the LoRA modules themselves.

### A.2 FEW-SHOT PLAYERS AND DATA IMBALANCE

While our dataset has a median game count of 3,479 games, many players may have as few as 10-50 games. Our evaluation of few-shot learning shows that 100 games is sufficient to learn the style vector of an unseen player. However, one might still ask how accurately such a style vector is given a very small number of games. To explore this, we first split a player into disjoint sets of 10, 25, 50, 100, 500, and 1,000 games. We then train a style vector on each set. As a baseline, we train a style vector on 10,000 games and track the cosine similarity of the smaller-set style vectors relative to this baseline vector. We show the results in Figure 7.

### A.3 INTERPOLATING OVER THE STYLE SPACE

To further investigate the effect of interpolation on style, we compare the playing style of two random players to a third player produced by averaging their style vectors. The result is shown in Figure 8, where as before, we use a set of human-interpretable evaluation metrics from Stockfish

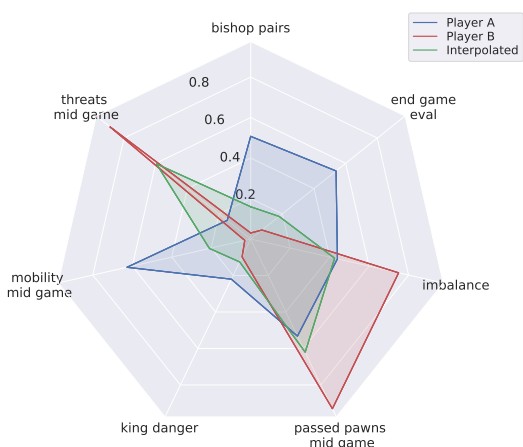

Figure 8: Player model strengths retrieved from Stockfish evaluations of two randomly sampled players compared to an intermediate player generated by averaging the source vectors.

to characterize each player's style. We observe that the green player (labeled Interpolated) retains an intermediate set of skills relative to the two source players. This corroborates the results of Figure 4, where we show that a style vector trained by merging two players' game sets has high cosine similarity to a style vector generated by averaging the style vectors trained on each game set individually.

## A.4 IMPLICIT STATIONARITY ASSUMPTIONS

Most of the existing work in chess assumes that a player remains stationary over time and across gameplay situations. However, in reality, a player's style may depend on the type of opponent they are facing, which opening is used, which stage of the game they are in (opening, middle, endgame), and so on. For instance, McIlroy-Young et al. (2021) observe that stylometry accuracy drops when removing the opening (e.g., the first 15 moves) moves, suggesting that the opening has an outsized effect on style identification. Our approach does not rely on these assumptions and can in principle be applied to arbitrary subsets of a player's data. For instance, one could split a player's data into opening, middlegame, and endgame moves and train a separate style vector for each. One could further split the data based on which defense the opponent uses, what time of the day it is, etc.. Despite treating players holistically and avoiding any splits of their data, we are still able to capture the peculiarities of each individual's playing style and perform stylometry with high accuracy. This also enables us to compare our results to those of prior work, which also treats player data holistically.

