# OpenReview forum: "Generative Modeling of Individual Behavior at Scale"
_ICLR.cc/2024/Conference — Submitted to ICLR 2024_

### Official Review · Reviewer_rXFi · 2023-10-31

**Soundness:** 3 good
**Presentation:** 2 fair
**Contribution:** 2 fair
**Rating:** 5
**Confidence:** 3

**Summary:**

This paper scales up the individual-level behavior simulation at scale using Parameter Efficient Fine-Tuning (specifically Poly LoRA). This has significantly scaled up the simulation, with the neat addition of new capabilities such as individual-style generation and style analysis. This work demonstrates how the latest PEFT methods can significantly scale up fine-grained human behavior simulation. The authors use chess game analysis as an example, but similar simulations could be possible in other domains.

**Strengths:**

* By combining multiple of the latest scalability fine tuning techniques, the authors were able to successfully scale up the individual-level simulation in unprecedented ways.
* Authors provide interesting observations from the simulation proving the individual models are indeed helpful for the scaled analysis of human behavior. e.g. Figure 5

**Weaknesses:**

* Besides the interesting adaptation in the fine-grained human behavior analysis, the actual technical contribution is quite limited because they are simple extensions of the previous work on the chess domain.
* Even considering limited work in the chess game simulation, the empirical comparison is pretty weak. They could have considered other baselines such as ones with standard DNNs for example from one model.
* Technical contribution to the machine learning communities seems to be limited since the work is a straightforward extension o f the Maia model with small modifications.

**Questions:**

Please take a look at the weakness section.

---

> ### Author Response · Authors · 2023-11-20
>
> Thank you for your valuable feedback, and for appreciating our scalable analysis of individual behavior. Although our model architecture builds on existing PEFT methods and the base Maia architecture, we believe our technical contributions go well beyond this architecture, as we explain in our general response above. We address the remaining points you raised below.
>
> ## On empirical comparisons and baselines:
>
> Although the set of comparisons made within our paper are limited, we believe they are sufficient to show the efficacy of our method. The models which we test against represent the state-of-the-art in their respective domains, and also in other domains. For example, we compare against individual model fine-tuning (McIlroy-Young et al., 2022) and to a Transformer-based method (McIlroy-Young et al., 2021); we achieve superior results to both, at much larger scale, while providing *generative* models. In their paper, McIlroy-Young et al. (2021) include comparisons of their Transformer encoder, which is a modified version of the Vision Transformer (ViT) architecture, to other architectures within the stylometry domain. The ViT architecture has been shown to perform at state-of-the-art levels in many other domains. Additionally, Maia is based on the Squeeze-and-Excitation Residual Network, which has been shown to perform well in image classification. Given the strength of these base models in both chess and other domains, and the fact that we compare against state-of-the-art methods that build upon them, we believe that adding weaker models would not yield significant added value to our analysis.
>
> However, it is possible that we are misunderstanding the comparison the reviewer had in mind; if the reviewer could clarify what they meant by standard DNN architecture, we would be happy to include this comparison in our evaluation.

---

### Official Review · Reviewer_XeAJ · 2023-11-01

**Soundness:** 3 good
**Presentation:** 2 fair
**Contribution:** 2 fair
**Rating:** 5
**Confidence:** 3

**Summary:**

This paper proposes a novel method to model individual behavioral stylometry. Specifically, the authors take it as a multi-task learning problem where each task refers to a distinct person. The method is parameter-efficient and thus can perform stylometry at an unprecedented scale with few-shot learning enabled. Experimental results on chess data show the effectiveness of the method.

**Strengths:**

1. The topic is interesting and the proposed way for modeling human individual behavior looks novel to me.

2. Overall the paper is clearly written and easy to follow.

**Weaknesses:**

1. The method looks to some extent incremental to me: some existing techniques such as Lora and Polytropon are combined and employed for a specific task.

2. Currently, the empirical result is only on a chess dataset. As the paper aims at modeling human behavior, it would be good if more related datasets could be considered. See question 4.

minors:

Typos: "we use this to to evaluate few-shot learning ..." and "we run a simulated tournament between the them".

**Questions:**

1. What is the intuition behind to model each player as a task?

2. Is there any data imbalance / long tail problem under this setting?

3. Can we have a fig5-like result for interpolation over the style space?

4. Can the method be applied to games with asymmetric information like poker?

---

> ### Author Response · Authors · 2023-11-20
>
> Thank you for your valuable feedback, and for appreciating the novel aspects of our approach. Although our work applies existing PEFT methods (e.g., Polytropon and MHR) to the domain of chess, our technical contributions go beyond this application, as we explain in our general response above. We address the remaining points you raised below.
>
> ## On the intuition behind modelling each player as a task:
>
> In multi-task learning, there is a trade-off between sharing information across (similar) tasks and differentiating between (dissimilar) tasks. Given any two tasks, some aspects will be similar and some will be different. We realized that the same is true of players in a game: many players share similar strategies or aspects of their playing style, but each player is still unique. Thus, we saw the potential of modeling shared characteristics across the players (e.g., through skill modules), while allowing them to express their peculiarities (e.g., through style vectors). This paradigm mapped well to recent PEFT approaches used in multi-task learning, which inspired us to make the conceptual connection between gaming and this field.
>
> ## On the issue of data imbalance and the long tail problem:
>
> Chess player data does have some significant data imbalance, but we do not believe this is a significant issue for our method given our few-shot learning results on <= 100 games. In Table 2, for example, we show that training on completely unseen players using just 100 games is enough to identify them from a set of 10,000 other players. Additionally, Figure 2 shows that our method can model a player with as few as 50 games and achieve nearly 60% move-matching accuracy. Finally, we include a chart linked below showing the cosine similarity of style vectors generated using different sizes of disjoint subsets of games, compared to a style vector trained on 10,000 games. Observe that even with just 10 games, the cosine similarity is 0.422, which is over two standard deviations (sd=0.103) above the average pairwise cosine similarity across the population (0.188). We have added some discussion regarding data imbalance and players with lower game counts to our paper.
>
> [Link to Figure] https://imgur.com/a/lqt9eiZ
>
>
> ## Figure 5-like chart for interpolation on the style space:
>
> This is a great question. Below is a link to the requested chart, where we produce a new player (labeled Interpolated) by averaging the style vectors of two random players from our dataset. While the interpolated player isn’t an exact average, it aligns quite well in most dimensions. We have added this example to our paper.
>
> [Link to Figure] https://imgur.com/a/NcxRQcI
>
> ## On applications to games with asymmetric information:
>
> This is an interesting question to explore. In principle our approach could extend to games with asymmetric information, because aspects of a player’s strategy or playing style can still be learned from their actions. When applying our approach to visual 1-vs-1 games, asymmetry arises due to partial observability of the state space, such as a player only seeing what is in their field of view. In poker, the current state would not include the specific cards held by other players. Nevertheless, such partially observed states can still be encoded and provided as input to the base model, atop which our adapter architecture is applied (see Fig. 1). Also see our general response for further comments on generalizing our work beyond the domain of chess.

---

> > ### Comment · Reviewer_XeAJ · 2023-11-21
> >
> > Thank you for your response. After reading all the review comments, I tend to keep my rating unchanged. The paper focuses on modeling individual behavior, which is itself a rather narrow domain. Therefore, it would be good if at least one more dataset could be included in the experiments.

---

### Official Review · Reviewer_RNKg · 2023-11-01

**Soundness:** 3 good
**Presentation:** 3 good
**Contribution:** 3 good
**Rating:** 6
**Confidence:** 2

**Summary:**

The paper proposes a method to model individual behavior in chess using a multi-task learning framework that leverages parameter-efficient fine-tuning (PEFT) methods. The paper introduces a style vector for each player that captures their distribution over latent skills learned from a shared inventory of adapters. The style vector enables generative modeling, stylometry, and style analysis and synthesis of players.
The paper evaluates the method on a large-scale dataset of over 47,000 players and 244 million games, and shows that it can perform stylometry with high accuracy, predict Elo ratings, probe player styles, and create novel human-like styles.

**Strengths:**

1. The paper proposes a novel and rigorous method to model individual behavior in chess using a multi-task learning framework that leverages parameter-efficient fine-tuning methods.
2. The paper introduces a style vector for each player that captures their distribution over latent skills learned from a shared inventory of adapters. The style vector enables generative modeling, stylometry, and style analysis and synthesis of players.
3. The paper evaluates the method on a large-scale dataset of over 47,000 players and 244 million games, and shows that it can perform stylometry with high accuracy, predict Elo ratings, probe player styles, and create novel human-like styles.

**Weaknesses:**

1. The paper focuses on modeling individual behavior in chess, which is a specific and narrow domain. The method may not generalize well to other domains or tasks that have different characteristics or constraints.
2. The paper assumes that the players’ styles are stationary and independent of the context or the opponent. However, in practice, players may adapt their styles to the situation or the opponent, which could affect the accuracy and validity of the method.

**Questions:**

How do you see this method to generalize beyond a game setting, e.g., to role-play text generation (e.g., Character AI)?

**Details Of Ethics Concerns:**

The paper does not address the ethical implications of using generative models to mimic human behavior, such as privacy, security, fairness, and accountability. The method could be used for malicious purposes, such as impersonation, fraud, manipulation, or surveillance, which could harm individuals or society.

---

> ### Author Response · Authors · 2023-11-20
>
> Thank you for your valuable feedback, and for appreciating the rigor in our work. We address your first question in the general response; we address your second point below.
>
>
> ## On implicit stationarity assumptions of our approach:
>
> It is true that our approach marginalizes over player opponents/contexts and a player’s evolution over time. This is a standard assumption (also made in previous work) which does not affect the validity of our method, in that it is valid to treat each player holistically and see how the method performs. In fact, we achieve strong results when viewing each player holistically: e.g., over 94% stylometry accuracy over a reference set of ~48K players. However, the reviewer is absolutely right that changing the granularity of our analysis—such as accounting for opponent style, in-game context, or a player’s temporal evolution—are possible with our method and depending on the downstream applications, could be desirable. Here are some relevant observations along these lines:
>
> - When excluding the opening part of the games (e.g., the first 15 moves) from our analysis, stylometry accuracy reduces, indicating that the opening has an outsized effect on style identification relative to the middlegame and endgame. This phenomenon was also observed by McIlroy-Young et al. (2021).
>
> - A few players in our dataset open a second account after a certain time period, creating a natural temporal split in their data. When comparing the style vectors across these time periods, we observe that they have very high cosine similarity and achieve almost identical move-matching accuracy on each other’s datasets. This is partly because most players exhibit stationary performance (in terms of Elo/playing strength) over time, and because some elements of style persist across playing strength increases.
>
> - One of our experiments (Fig. 4) merges the datasets of two random players—in effect forcing a non-stationarity—and compares the resulting style vector to the arithmetic mean of the individual style vectors, showing they have high cosine similarity.
>
>
> In general, we can segment our dataset in a variety of ways and readily apply our method to train a separate style vector for each segment. *If the reviewer has a particular segmentation in mind, we would be happy to implement it*. We have also added this discussion to our paper.

---

### Author Response · Authors · 2023-11-20
**General Response**

Dear reviewers, thank you very much for your valuable feedback! We appreciate the time you spent providing us with constructive advice to improve our submission. We list below our general response and address individual concerns in separate threads.


## On the generalizability of our approach beyond chess:

While our experiments are centered around chess, we believe that it provides an interesting and powerful research environment for several reasons: (1) there is a vast amount of human data across tens of thousands of unique players; (2) player behavior is highly diverse and multimodal (see Fig. 5); and (3) there exist human-interpretable heuristics for evaluating actions and positions that allow us to analyze player styles. Together, these properties enable us to explore stylometry and generative modeling at the individual level and at unprecedented scale.

Given that the original parameter-efficient fine-tuning (PEFT) work underlying our approach was applied to natural language tasks, in principle our approach could extend to such settings, including character-based text generation. Similarly, our approach could in principle extend to other turn-based or 1-vs-1 graphical games. We agree with the reviewers that it would be interesting to explore these applications beyond chess. However, given the richness and complexity arising within the chess domain and our method of modeling individual behavior as “tasks” in a multi-task learning framework, we believe the scope of the current submission is appropriate and hope to explore other applications in future work.


## On the novelty and technical contributions of our work:

Although our work builds on existing PEFT methods (e.g., Polytropon and MHR) and the Maia chess architecture, we believe our technical contributions go well beyond a simple application of existing methods. Specifically, we contribute in the following ways:

**Conceptual.** By viewing behavioral modeling as a multi-task learning problem, where each person is represented by a “task”, we build a conceptual connection between these two areas that enables a novel application of PEFT methods at massive scale (e.g., as many unique players as there are in a game).

**Scale.** Currently, modeling individual behavior in games requires creating a separate model for each player. This does not scale beyond a few hundred players, since fine-tuning individual models is computationally expensive. For instance, creating a model per player using the method of McIlroy-Young et al. (2022) takes an average of 20 minutes per model on an A100 80GB GPU. This limits the total throughput of the system to 72 models per GPU daily, or about 500 models in a week. Our method is able to model 47,864 players on just one A100 80GB GPU within one week.

Most of the current work on PEFT in multi-task learning also does not test methods at scale. For instance, Polytropon (Ponti et al., 2022) tests on the order of a hundred tasks. Our method shows that these techniques, when applied to modeling individual player behavior, can scale to tens of thousands of tasks, which is several orders of magnitude more than previous work. We believe this demonstration of scale is a worthwhile contribution.

**Improved capabilities.** Compared to the work on Maia, our approach offers significant improvements in capability and efficiency for behavioral cloning and stylometry. To do stylometry with the base Maia model, each player’s set of position-move pairs must be compared to the output of every model in the reference player set, which is very computationally inefficient, as McIlroy-Young et al. (2021) also observe. In contrast, our method requires only computing cosine similarity across the set of candidate player vectors, which is very fast. We also achieve comparable accuracy to a Transformer encoder in stylometry, *while being able to generate actions in the style of each player*, which the Transformer-based method cannot do. In other words, our scalable stylometry method also yields a scalable behavior cloning method.

**Novel analyses.** Our method allows for novel analyses which were not previously possible. We are able to generate new styles by taking advantage of the latent space induced by the learned style vectors—e.g., by averaging the style vectors of two players (Fig. 4, Fig. 6). Since our approach is generative, we can observe how these new players behave and evaluate their decisions using human-interpretable concepts (Fig. 5, and our new analysis for Reviewer XeAJ in https://imgur.com/a/NcxRQcI). This is not possible with individual model fine-tuning or Transformer-based embeddings that are not generative.


In summary, we believe that these improvements in scalability, capability, and analysis are significant technical contributions which go beyond a simple extension of previous work. Additionally, we hope our conceptual connection between multi-task learning and individual behavior modeling inspires future work.

---

### Meta-Review · Area_Chair_7xC4 · 2023-12-05

**Metareview:**

This paper proposes a new personalized architecture for predicting and generating moves of specific chess players. The empirical results suggest that the new approach improves the accuracy in move prediction by a few percent over the baseline. However, the reviewers had concerns regarding the contributions of the paper, given the existing literature on personalization and adaptation. While the results in chess can be significant by themselves, the authors should perhaps consider extending the paper and demonstrating more impressive applications using this model. By itself, the technical contributions on the specific adaptation methods and the empirical gains from using it are too limited.

**Justification For Why Not Higher Score:**

While building personalized chess models is novel and could make great impact, the technical and empirical contributions of the work are currently too narrow.

**Justification For Why Not Lower Score:**

n/a

---

### Decision · Program_Chairs · 2024-01-16

Reject